# Well-NeRF: Ensuring Well-Posed Neural Radiance Fields via View Frustum and Shadow Zone Based Regularization

## Abstract

Neural Radiation Field (NeRF) often produces many artifacts with sparse inputs. These artifacts are primarily caused by learning in regions where position inference is not feasible. We assume that the main cause of this problem is the incorrect setting of boundary conditions in the learning space. To address this issue, we propose a new regularization method based on two key assumptions: (1) the position of density and color cannot be inferred in regions where the view frustum does not intersect, and (2) information inside opaque surfaces cannot be observed and inferred, and thus cannot contribute to the rendering of the image. Our method aims to transform the NeRF model into a well-posed problem by regularizing learning in regions where position inference is not possible, allowing the network to converge meaningfully. Our approach does not require scene-specific optimization and focuses on regions where position inference is not possible, thereby avoiding degradation of model performance in main regions. Experimental results demonstrate the effectiveness of our method in addressing the sparse input problem, showing outstanding performance on the Blender synthetic datasets. Our method is designed to integrate seamlessly with existing techniques, providing an effective solution for sparse input scenarios and offering a foundational approach that serves as the first clue in addressing sparse input problems.

## 1 Introduction

Recently, the effectiveness of NeRF Mildenhall et al. (2021) has been witnessed in inferring scenes from unobserved perspectives by training on multiple photographic inputs. However, existing NeRF models often generate unrealistic scenes that are difficult for humans to comprehend. This problem is less prominent when the input data is dense but becomes more noticeable when the input data is sparse.

The "Near-far plane" parameter is an important but often unaddressed issue in many experiments, where it is frequently precisely tuned during experiments but not explicitly mentioned in previous papers. If this parameter is not accurately tuned for the region of interest (ROI), especially in cases of sparse input, it can result in poor object generation or numerous artifacts. This limitation is a critical issue that undermines the practical usefulness of experimental results. We interpret the improper tuning of the near-far threshold not only as a failure in setting training parameters, but also as a misconfiguration of the "spatial boundary conditions" during training.

"Proper boundary conditions" are essential for solving the problem effectively. As shown in Fig.1, improper boundary conditions result in an ill-posed problem. To address this, our model is designed to transform NeRF training into a well-posed problem, which we have named **Well-NeRF**.

An "ill-posed problem" occurs when learning is permitted in regions where position inference is not possible. In this situation, the NeRF training process may offer the model simpler but inaccurate solutions, such as increasing the weight in regions like the red areas shown in Fig.1. However, the result may deviate significantly from our intuition about the structure of the object, leading to critical errors.

Figure 1: **Ill-posed problems caused by conventional NeRF models**. (a) In the sparse input view, the region labeled with red color indicates where the view frustum does not overlap ($S = 1$, as shown in Equation equation 2). Position inference is not feasible in this area due to the ill-posed problem. (b) If the positional inference domain is not adequately constrained, the model may converge on an easier solution rather than generating a correct representation of the object. Consequently, the model may produce artifacts in ill-posed areas (such as in front of or behind the object). While these results may seem correct from the input view perspective, their inaccuracies become apparent as soon as the viewpoint shifts even slightly. (c) Ill-trained methods produce artifacts that are more dispersed along axes parallel to the visual field. The fragmentation of input view directions in reconstructed 3D models is a common phenomenon observed across various models under sparse data conditions.

We have developed an improved method for setting boundary conditions. Even with a finely tuned near-far threshold, there are limitations in accurately defining the boundary conditions of the training space, as it cannot clearly define the space where position inference is feasible.

We present two propositions that are true in order to develop a solution. First, in the non-overlapping parts of the view frustum, we cannot infer density and color according to position. Second, the information inside the opaque surface (shadow zone) cannot be observed and inferred, and cannot contribute to the rendering of the image. Based on this, we propose two regularization methods.

- Frustum score based regularization: This method calculates how much the view frustums overlap at the positions of points sampled in space and reflects this in the training process. If the view frustums do not overlap ($S = 1$, as shown in Equation equation 2), learning is heavily constrained. However, when two or more frustums overlap ($S \geq 2$), learning is allowed, and the gradient is scaled based on the degree of overlap, allowing different gradients to be applied depending on the extent of the overlap.
- Shadow zone regularization: This method reasonably approximates the color of the back of an opaque surface that cannot be learned. By constraining the complexity of the back of an opaque surface, the method allows the NeRF network to converge on a meaningful structure for the object.

Since this method automatically adjusts the training space based on the input data, no additional parameter tuning is required. Our method improves the NeRF model by limiting learning in areas where ill-posed problems can occur, ensuring that the model focuses on well-posed problems. Our approach works outside the main learning region of the model, effectively maintaining NeRF's core performance and potential. Furthermore, our method is built on top of the integrated model Nerfacto Tancik et al. (2023), making it easy to combine with other methods.

The goal of our model is to provide key insights and an initial step toward enabling various models to function effectively even with sparse input.

## 2 RELATED WORKS

### 2.1 NEURAL RADIANCE FIELD

Neural Radiance Fields (NeRF) Mildenhall et al. (2021) trains a deep neural network from a set of photos to represent the continuous 3D scenes, and uses volume rendering to render the scene. Since NeRF was first released to the public, various models Barron et al. (2021; 2022); Müller et al.

(2022); Fridovich-Keil et al. (2022); Wang et al. (2021) have evolved to overcome the shortcomings of the original NeRF. NeRF-W Martin-Brualla et al. (2021) improved the model to be trained on a variety of data under different conditions of contrast, brightness, and other factors. The problem of generating excessive noise for scenes with moving objects has also been solved by certain methods Pumarola et al. (2021); Park et al. (2021a;b). Other methods Wang et al. (2021); Jeong et al. (2021); Lin et al. (2021); Bian et al. (2023); Park et al. (2023) optimized the camera parameters of the input image to solve the problem of reliance on camera calibration techniques. Mip-NeRF Barron et al. (2021) improved rendering quality through an anti-aliasing method, while Mip-NeRF360 Barron et al. (2022) used scene contraction to achieve efficient modeling in unbounded scenes. Instant-NGP Müller et al. (2022) significantly accelerated training by using a hash encoding method. Nerfacto Tancik et al. (2023) model integrated various methodologies and improved rendering performance.

## 2.2 Neural Rendering With Sparse Dataset

Conventional NeRF-based methods face rendering performance issues and noise artifacts with sparse datasets. Although various models have been proposed to address these issues, limitations still exist. For practical use, it is necessary to effectively render continuous scenes with only a few datasets.

Methods such as Pixel-NeRF Yu et al. (2021) and Diet-NeRF Jain et al. (2021) required pre-training to achieve high performance. However, this pre-training process can introduce unintended new structures or artifacts that do not exist in the real view. Sparse-NeRF Wang et al. (2023) and DS-NeRF Deng et al. (2022) relied on additional prior knowledge of the depth value of the input scene. Obtaining this information typically requires additional sensors during image capture or separate depth estimation algorithms. These requirements for this additional information limit the versatility of the model.

Some methods have developed regularization techniques based on a cognitive view of geometric and visual shapes. Reg-NeRF Niemeyer et al. (2022) applied regularization for geometric information and appearance of the scene based on patches. However, this approach can induce additional learning costs and suffer from slow convergence problems. FreeNeRF Yang et al. (2023) improved rendering performance for few-shot dataset using a frequency regularization method that sequentially trains from low to high frequencies of positional-encoding. Additionally, it employed an occlusion regularization method, which could push back artifacts that are close to camera. However, these methods require optimized parameter settings for each scene, necessitating recursive experiments for successful rendering. Regularization techniques designed based on an intuitive approach to 3D geometric shapes or artifacts might work well in certain cases, but they had the potential to cause irrational or over-regularization in certain scenes.

Furthermore, a common characteristic among these models is the precise setting of a near-far threshold for the view frustum (ray). A near-far threshold that is precisely set to the region where the object exists effectively helps learn 3D geometric structure. However, setting this threshold as a hyperparameter for each dataset is not always feasible in real-world scenarios. Our method, which does not require this kind of heuristic hyperparameter value, is a highly versatile model that can be applied to various datasets.

## 2.3 Artifacts Removal

Many models aiming to address the sparse input problem in NeRF attempt to remove artifacts that obstruct the field of view. Floaters-no-more Philip & Deschaintre (2023) assumed that the scene near the camera undergoes frequent updates due to the large number of samples per space along the rays. Thus, it introduced a gradient scaling technique to reduce the learning rate for nearby distances, thereby mitigating the influence of proximity samples and eliminating artifacts. In addition, NeRFbusters Warburg* et al. (2023) used strategies to clean up the field of view by sampling points outside of the view frustum using random sampling and applying a regularization term to decrease density. However, these models are inherently designed for dense inputs and have limitations with sparse input datasets.

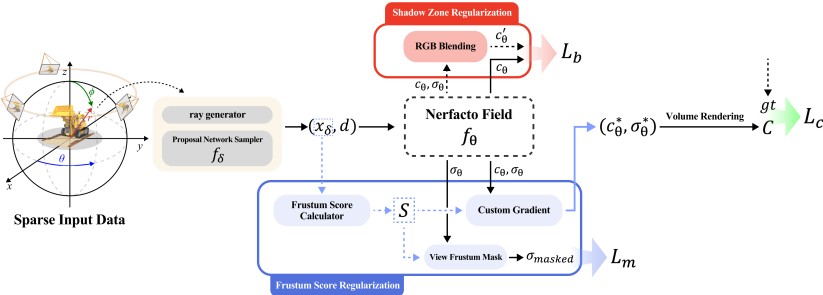

Figure 2: **Overall pipeline of the proposed method.** Our model is based on Nerfacto Tancik et al. (2023) and uses two regularization process. The solid line represents the path where parameters are updated by back-propagation. Output values of the Nerfacto Field, $c_\theta$, $\sigma_\theta$, are utilized in our regularization method. Frustum Score Regularization uses the position $x_\delta$ extracted from Proposal Network Sampler $f_\delta$ as input, yielding a score $S$ through Frustration Score Calculator. This score is utilized by the Custom Gradient and View Frustum Mask. The estimated color value of the 3D point, $c_\theta^*$, $\sigma_\theta^*$, generates the final color value through volume rendering. All three loss values are used for the final loss in equation 9 for the training.

## 3 PROPOSED METHOD

Sections 3.2 and 3.3 describe the proposed frustum score based regularization and shadow zone regularization, respectively. Fig.2 describes the overall pipeline of the proposed method.

### 3.1 PRELIMINARIES

#### 3.1.1 NEURAL RADIANCE FIELDS

(NeRF) utilize a deep learning framework to reconstruct three-dimensional scenes from a set of 2D images and synthesize novel viewpoints. The model receives spatial coordinates $\mathbf{x}$ and directional vectors $\mathbf{d}$ as inputs, and through a neural network parameterized by $\theta$, denoted $f_\theta$, it outputs predicted color $c_\theta$ and density $\sigma_\theta$. This relationship is encapsulated in the following equation:

$$(c_\theta, \sigma_\theta) = f_\theta(\mathbf{x}, \mathbf{d}). \tag{1}$$

Volume rendering utilizes the predicted color and density to calculate the transmittance along each ray, integrating these properties to synthesize the final color of the image. This process effectively compiles the contributions of color and density across the depth of the scene, producing photorealistic outputs.

#### 3.1.2 NERFACTO.

The Nerfacto model integrates various methodological advancements since the NeRF method has been introduced. Specifically, it combines techniques introduced in Mip-NeRF 360 Barron et al. (2022) (scene contraction, proposal MLP) and Instant-NGP Müller et al. (2022) (multi resolution hash encoding, spherical harmonics encoding). It extends the existing framework of NeRF and incorporates a number of recent methods to achieve improved visual accuracy and computational efficiency.

### 3.2 FRUSTUM SCORE REGULARIZATION

In regions where the View Frustum does not intersect at all(region observed from only a single input view), three-dimensional positional inference is fundamentally impossible. In such regions, positional color and density cannot be inferred from the input data. The color and density depend only on the characteristics of the neural network. This results in arbitrarily determined spatial information, which is an ill-posed problem for spatial information, as shown in Fig.1. Areas where this problem occurs can produce artifacts that are inconsistent with our understanding of the object. Therefore, it is important to determine if view frustums intersect.

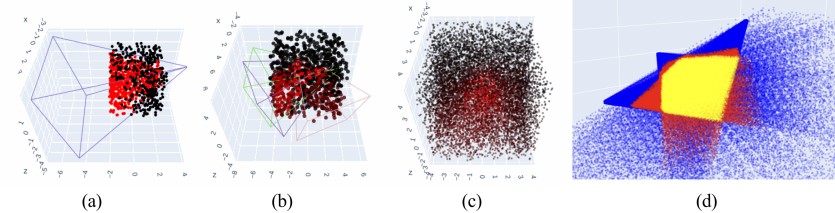

$$(a) \qquad\qquad (b) \qquad\qquad (c) \qquad\qquad (d)$$

Figure 3: **Frustum scores**. (a), (b), and (c) show frustum scores with random points. (a) shows that there is a single viewpoint, where red points represent that the position of the point is inside the frustum, while black points represent positions outside the frustum. Similarly, (b) indicates three viewpoints, with the closeness of the dot to red indicating a higher number of overlapping frustum regions in the position, and (c) indicates 100 viewpoints. (d) shows sample points in the frustum viewpoint of three.

### 3.2.1 FRUSTUM SCORE CALCULATOR

The Frustum Score Calculator (FSC) in equation 2 evaluates whether the points($\mathbf{x}$) sampled during the training process of the NeRF model are included in the view frustum of the input data. If there are $N$ image-transform matrix pairs of data with $N$ viewpoints, the score of each point is distributed from a minimum of 1 to a maximum of $N$. This score indicates how much of the view frustum each sampling point is included in, which reflects the confidence in the spatial information inferred from that point. Through the application of the FSC, the importance of spatial information as view frustums intersect can be taken into account during the training process. This contributes to improving the accuracy and efficiency of the NeRF model.

$$S = \text{FSC}(\mathbf{x}) = \sum_{j=0}^{n-1} \texttt{isInside}_j(\mathbf{x}), \tag{2}$$

Where j represents the j-th view frustum, $\mathbf{S}$ is the Frustum Score, and $\mathbf{x}$ is the input sampling point. The definition of the "isInside" function is described in detail in the supplementary materials.

### 3.2.2 VIEW FRUSTUM MASK.

Learning should be strongly constrained because positional inference is not possible on the regions with a frustum score of 1 (where the views frustum do not intersect). Thus, we mask the regions with a frustum score of 1 using $\sigma_{\mathbf{masked}}$ in equation 3 and enforce regularization such that sigma is zero based on the loss function $\mathcal{L}_{\mathrm{m}}$ in equation 4.

$$\sigma_{\mathbf{masked}} = \sigma \odot \mathbf{M} \text{ and } \mathbf{M}(i) = \begin{cases} 1, & \text{if } score(i) < 2 \\ 0, & \text{otherwise.} \end{cases} \tag{3}$$

$$\mathcal{L}_{\mathrm{m}} = \text{MSE}(\sigma_{\mathbf{masked}}, \mathbf{0}). \tag{4}$$

### 3.2.3 CUSTOM GRADIENT FUNCTION.

Positional inference is possible in regions with a frustum score of 2 or more. However, due to the occurrence of shadow zones, inaccurate positional inference may occur even in this region. To alleviate this phenomenon, we apply a gradient difference in backpropagation so that regions with more intersecting view frustum can be learned priorly. Within the NeRF model, utilizing the score($\mathbf{S}$) obtained through the Frustum Score Calculator in equation 2, we designed a custom gradient function $f_{cg}$ in equation 5 and equation 6 to ensure that $\mathbf{c}_\theta$(color) and $\sigma_\theta$(density) value generated from 3D coordinate $\mathbf{x}$ through $F_\theta$(Nerfacto Field) carry a gradient proportional to the score during back-propagation.

$$f_{cg,\text{Forward}}(\mathbf{c}_\theta, \sigma_\theta, \mathbf{S}) = (\mathbf{c}_\theta^*, \sigma_\theta^*), \tag{5}$$

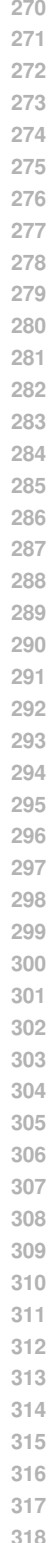
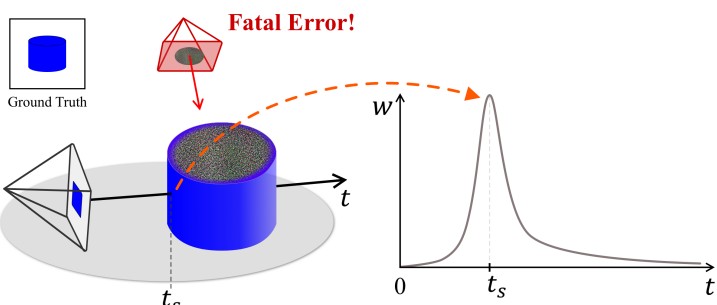

Figure 4: **Illustration of shadow zone**. When observing an opaque object from a particular view-point, the final color value is primarily determined by the color value of the object's surface. As the ray's depth $t$ increases, the weight value rapidly increases and decreases near the object's surface, $t_s$, as shown in the graph. Undeterminable information inside an object can cause learning instability and optical illusions when viewed from different perspectives.

$$f_{cg,\text{Backward}}\left(\frac{\partial L}{\partial \mathbf{c}}, \frac{\partial L}{\partial \sigma}\right) = \left(\frac{\partial L}{\partial \mathbf{c}} \odot \mathbf{S_{norm}}^2, \frac{\partial L}{\partial \sigma} \odot \mathbf{S_{norm}}^2\right), \quad (6)$$

where $\mathbf{S_{norm}}$ is the normalized score and $L$ represents the input for the back-propagation gradient.

### 3.3 SHADOW ZONE REGULARIZATION

Information behind opaque surfaces with low transmittance is not observable. Therefore, information observed at a particular view frustum cannot contribute to learning behind opaque surfaces. Therefore, learning behind opaque surfaces may introduce unnecessary complexity to the model without contributing to image rendering. Based on this finding, we develop a regularization method that can reduce the complexity of RGB colors behind opaque surfaces. This implements boundary conditions based on object surfaces.

#### 3.3.1 RGB BLENDING

In the NeRF model, the color rendered by ray is largely determined by the surface of the opaque object (the high weight points close to the origin of the ray). As training progresses, it becomes increasingly likely that the color values at the back of opaque object surfaces are determined not by input data, but solely by network characteristics, resulting in noisy values, as in Fig.4.

To address this issue, we develop a regularization method based on the loss function $\mathcal{L}_b$ in equation 7, which gradually blends the color of the opaque surface into the interior of the object, ensuring a more stable determination of interior object color. This constraint is computed via a self-feedback mechanism, requiring no additional data input. Moreover, since it operates within a single ray, it can be seamlessly integrated without altering the existing random ray sampling scheme. This method reduces meaningless network complexity and helps the model converge to more meaningful values.

$$\mathcal{L}_b = \text{MSELoss}(C_i', C_i), \quad (7)$$

where $C_i' = (C_i w_i + C_{i-1} w_{i-1})/(w_i + w_{i-1})$ and the index $i$ represents the i-th sampling point along the ray. The color loss can be calculated as follows.

$$\mathcal{L}_c = \text{MSELoss}(\text{GT}, C), \quad (8)$$

where $C = \sum_i (C_i w_i)$. Then, the total loss can be designed by combining $\mathcal{L}_b$ in equation 7 with the color loss $\mathcal{L}_c$ in equation 8 and the view frustum loss function $\mathcal{L}_m$ in equation 4.

$$\mathcal{L}_{Total} = \mathcal{L}_c + \mathcal{L}_m + \lambda \mathcal{L}_b, \quad (9)$$

$$\lambda = clone(\mathcal{L}_c). \quad (10)$$

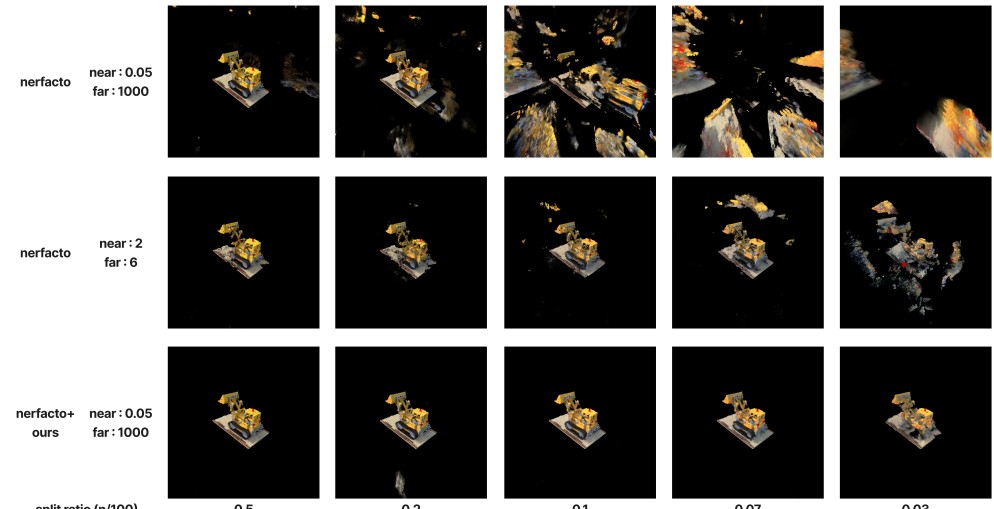

Figure 5: **Qualitative results in scenarios with sparse inputs.** We demonstrate that when the near-far threshold is set under broad conditions, severe fragmentation artifacts, as mentioned in Figure 1 are observed (1st row). Additionally, we observe that simply adjusting the near-far threshold, without any modification to the model, significantly mitigates this issue (2nd row). Our method shows that this problem is almost completely controlled (3rd row). Bottom value represents the count ratio of Blender training dataset. For example, a value of 0.1 indicates the use of 10 datasets as input (assuming the total count of Blender training datasets is 100).

The clone value of $\mathcal{L}_c$, detached from the computation graph, is used as $\lambda$ to facilitate initial convergence. As learning progresses and $\mathcal{L}_c$ decreases, the regularization term is applied less.

## 4 EXPERIMENTS

Our experiments proceed as follows. First, we demonstrate that setting incorrect boundary conditions in the learning space under sparse data conditions leads to critical errors. We also show that previous models addressing the sparse data problem have overlooked this issue. Next, we present how our model effectively resolves this problem.

### 4.1 EXPERIMENTAL SETTINGS

**Datasets** & **Metrics.** Using Synthetic Blender data, we conducted a comparison between our method and FreeNeRF using the same setup (8 view training and 25 view evaluation). Additionally, the image resolution was downsampled by 2x. For the other experiments, we utilized the 100 training views provided by NeRF Mildenhall et al. (2021) and divided them using the "Train split fraction" function of NerfStudioTancik et al. (2023). Our quantitative analysis includes PSNR, SSIM, and LPIPS metrics. We also conducted experiments using the DTU LLFF dataset, a real-world photography dataset, following the training and evaluation protocols of FreeNeRF. Detailed descriptions and results can be found in the supplementary materials.

| Configuration | PSNR | SSIM | LPIPS |
|:---:|:---:|:---:|:---:|
| Ours | 29.65 | 0.933 | 0.066 |
| Nerfacto | 28.51 | 0.909 | 0.101 |

Table 1: **Performance Comparison with dense input data:** Comparison of average metrics between Nerfacto and our model for Blender synthetic objects with 100-view training input. Our model typically enhances performance even in the dense input conditions where NeRF is utilized. This implies a lower risk of potential performance decline due to over regularization. This is an important outcome for the application of our method in integrated models

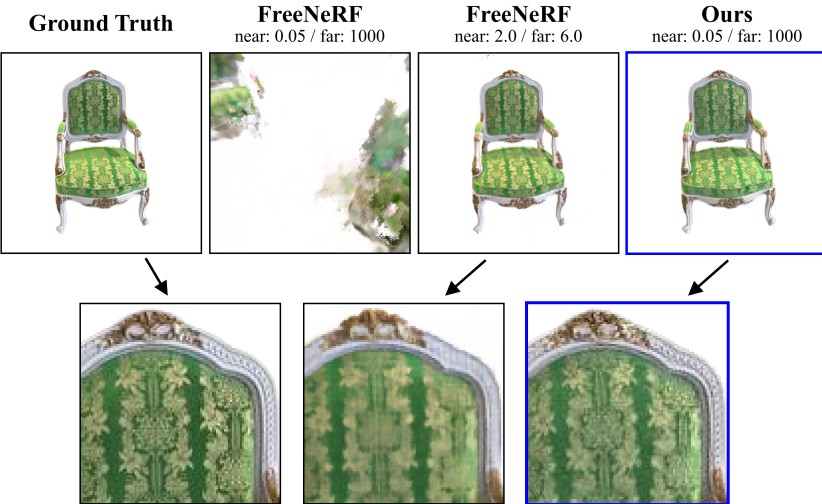

Figure 6: **Qualitative comparison with FreeNeRF Yang et al. (2023).** Our method shows outstanding performance for sparse blender dataset even with no near-far threshold compared to FreeNeRF Yang et al. (2023) (Blender synthetic objects and 8 input views).

**Implementation Details** Our method was implemented based on Nerfacto v1.0.0. While we utilized the core methods of Nerfacto, including proposal sampler, scene contraction, multi-resolution hash encoding, spherical harmonics encoding, and volume rendering, we disabled additional methods such as appearance embedding, distortion loss, and predict normal. However, our method has been designed to function seamlessly even with these additional methods enabled. Please refer to the supplementary material for further details. Our model maintains Nerfacto's speed and accuracy while solving the sparse input issue. It trains in about 10 minutes with 30k iterations on a single RTX 4090 GPU.

**Hyper-Parameters** We used the default hyper-parameters from Nerfacto Tancik et al. (2023).The near-far plane in Nerfacto is set to [0.05-1000] to cover a wide range of scenes. Our method was implemented to work with hyperparameters Free, so no additional parameter settings were required.

## 4.2 COMPARISON WITH OTHER METHODS

Table 2 quantitatively compares our model with recently developed models that do not require pretraining. In particular, FreeNeRF achieves high performance only when the near-far plane of the view frustum is precisely set around the region of interest (ROI). While this approach provides optimized results for a particular scene, it limits the generality of the model. We observed that the performance of FreeNeRF is significantly lower when the near-far plane is set wide. The model used as the backbone in this study also achieved a significant performance improvement when the near-far plane is set precisely However, accurately setting the near-far plane becomes challenging in the absence of detailed information about the acquisition equipment, method, and target object. Failure to mention these factors may hinder the proper evaluation of the model. In contrast, our proposed model was designed with the goal of reliably performing across a wide range of near-far

| Configuration | PSNR | SSIM | LIPS |
|---|---|---|---|
| Base Nerfacto (near, far opt) | 21.48 | 0.797 | 0.228 |
| Base Nerfacto (near,far **broad**) | 13.97 | 0.614 | 0.481 |
| FreeNeRF (near, far opt)Yang et al. (2023) | 24.25 | 0.883 | 0.098 |
| FreeNeRF (near, far **broad**) | 10.41 | 0.657 | 0.367 |
| Base Nerfacto + Ours (near,far **broad**) | 24.37 | 0.866 | 0.105 |

Table 2: **Comparison of our model with the baseline model (Nerfacto) and FreeNeRF:** Evaluation of the average metric of Blender synthetic objects. The yellow highlight indicates experiments under broad near-far [0.05,1000] plane conditions, with 'opt' meaning that the near-far is set to [2,6].

|  | chair | drum | ficus | hotdog | lego | materials | mic | ship | Average |
|---|---|---|---|---|---|---|---|---|---|
| Nerfacto | 13.00 | 12.04 | 16.83 | 13.53 | 11.40 | 13.05 | 24.50 | 15.56 | 14.99 |
| +fsm | 19.89 | 14.24 | 23.60 | 20.14 | 21.89 | 21.59 | 29.15 | 18.44 | 21.12 |
| +fsm+fsgs | 21.89 | 19.33 | 25.46 | 20.97 | 23.60 | 22.28 | 30.23 | 20.00 | 22.97 |
| +fsm+fsgs+sz | 24.79 | 21.51 | 25.19 | 25.67 | 24.63 | 21.70 | 29.27 | 21.04 | 24.23 |

Table 3: **Quantitative comparison for ablation study on our methods** in terms of PSNR. fsm: Frustum Score Mask, fsgs: Frustum Score Gradient Scale, sz: Shadow Zone (Blender synthetic objects and 8 input views).

plane settings (0.05 to 1000), and we have achieved this objective. This suggests that the model has the potential to be universally applicable across a variety of environments and conditions. Table 1 shows comparison of average metrics between Nerfacto and our model for Blender synthetic objects with 100-view training input. Our model typically enhances performance even in the dense input conditions where NeRF is utilized.

Fig.5 qualitatively compares our model with the Nerfacto model using the sparse dataset. As depicted in the figure, our model demonstrates significant superiority over the Nerfacto model in scenarios with sparse inputs. As shown in Fig.6, our method shows outstanding performance for sparse blender dataset even with no near-far threshold compared to FreeNeRF Yang et al. (2023).

### 4.3 ABLATION STUDY

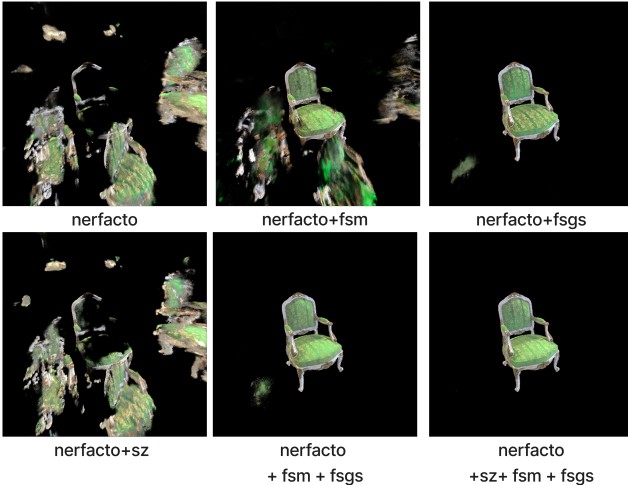

Figure 7: **Qualitative comparison for ablation study.** fsm: Frustum Score Mask, fsgs: Frustum Score Gradient Scale, sz: Shadow Zone Regularization (Blender synthetic objects and 8 input views).

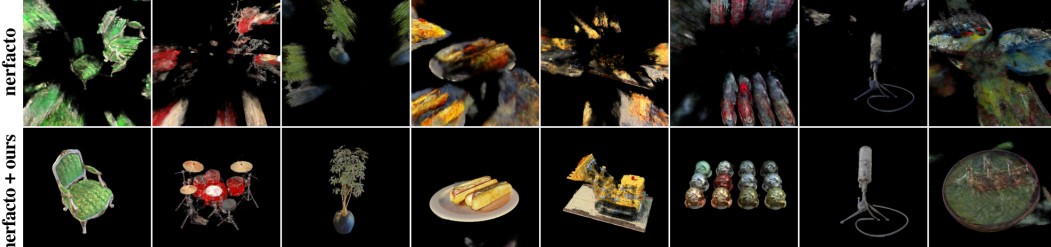

Figure 8: **Qualitative comparison for ablation study.** Qualitative results for a wider range of classes compared to those in Fig.7. Comparison between "Nerfacto" model and "Nerfacto + ours(sz+ fsm+ fsgs)" (Blender synthetic objects and 8 input views).

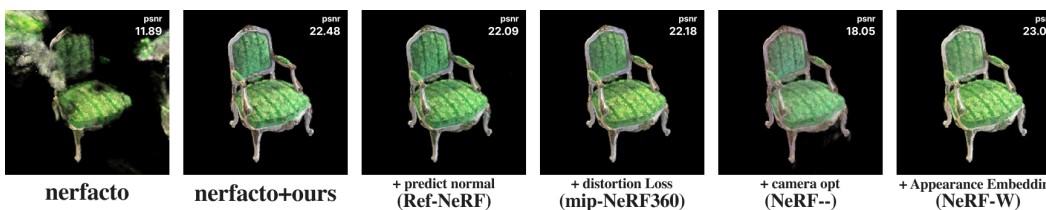

| nerfacto | nerfacto+ours | + predict normal (Ref-NeRF) | + distortion Loss (mip-NeRF360) | + camera opt (NeRF--) | + Appearance Embedding (NeRF-W) |

Figure 9: **Experimental results combining additional methods.** Our proposal is designed to seamlessly integrate with other methods, making it easy to assess the impact of combining our approach with additional techniques for sparse data problems (Blender synthetic objects and 8 input views).

Table 3 shows clear quantitative improvements when our proposed method is applied incrementally. Qualitatively, as shown in Figs.7 and 8, when using Nerfacto alone, the object does not converge in the region we predict. This is because converging in areas where the view frustum does not intersect (in ill-posed areas) is an easier solution for the NeRF model. When all the methods are combined, we observe the cleanest convergence.

## 5 CONCLUSION

In this paper, we presented Well-NeRF, a universal solution to the sparse input problem in NeRF. The key to our success lies in the foundation of our implementation, which starts from two true propositions, enabling our method to have fewer counterexamples. Since NeRF was first introduced, it has undergone significant developments, and efforts to integrate solutions for different problems continue. However, a clear integration addressing the sparse input problem has not yet been proposed, largely because of the existence of incompatible model structures or the potential for degrading the base model's performance. Our model does not compete with the base model, as it operates in different domains without leading to negative interactions, thereby not degrading the performance of the base model. Our model, developed on an integrated model framework, easily combines with various models without conflict, as shown in Fig.9 Our proposal has some limitations. However, applying the 'definition of the region where positional inference is possible' as spatial boundary conditions in a NeRF model to create a well-posed problem has clearly shown both qualitative and quantitative effectiveness in sparse problems. We hope our approach contributes to the advancement of diverse NeRF models, making them capable of addressing the sparse input problem.

## 6 LIMITATION

Distant backgrounds are difficult for our model to handle because it is nearly impossible to ensure overlap in the view frustum under sparse data conditions. Therefore, we conducted experiments in a limited manner for data captured under such conditions. The results can be found in the supplementary materials.

Frustum Score Calculator can be used to determine the most basic ill-posed areas. However, determining ill-posed areas becomes more complicated when there are objects and walls in a large complex structure. For example, the area behind a wall should not be counted even if it is inside the view frustum. Our shadow zone regularization method partially solves this problem, but it is not yet complete. In further study, we will develop a model that can efficiently determine ill-posed areas in various situations.

Our method does not completely control optical illusion artifacts. The artifacts we do not control occur in regions where inference is possible ($S \geq 2$, as shown in equation 2) and where multiple solutions may exist. Determining whether this is a true value or an optical illusion is not possible with the basic Nerf model. Additional regularization is required to address this. However, regularization designed with human intuitive assumptions about appearance and geometry is likely only applicable to certain scenes. Therefore, even if it improves performance in some scenes, it is likely to have a side effect on scenes in general. Therefore, we need to develop regularization methods that satisfy most of the human intuition and do not degrade NeRF learning.

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

## A    APPENDIX

### A.1    EXPERIMENTAL RESULTS ON DTU AND LLFF DATASETS

| Configuration | PSNR | SSIM | LPIPS |
|---|---|---|---|
| FreeNeRF (near,far broad) | 9.019 | 0.301 | 0.607 |
| Base Nerfacto + Ours (near,far broad) | 22.12 | 0.804 | 0.185 |

Table 4: **Comparison of our model and FreeNeRF DTU 9 input views.** Evaluation of the average metric of the DTU dataset, with near and far values set to a broad range [0.05,1000]. It shows stable convergence and high performance compared to FreeNeRF over a wide range of nearand far settings.

| Configuration | PSNR | SSIM | LPIPS |
|---|---|---|---|
| FreeNeRF (near,far broad) | 12.34 | 0.209 | 0.641 |
| Base Nerfacto + Ours (near,far broad) | 19.68 | 0.687 | 0.215 |

Table 5: **Comparison of our model and FreeNeRF LLFF 6 input views.** Evaluation of the average metric of the LLFF dataset, with near and far values set to a broad range [0.05,1000]. It shows stable convergence and high performance compared to FreeNeRF over a wide range of near and far settings.

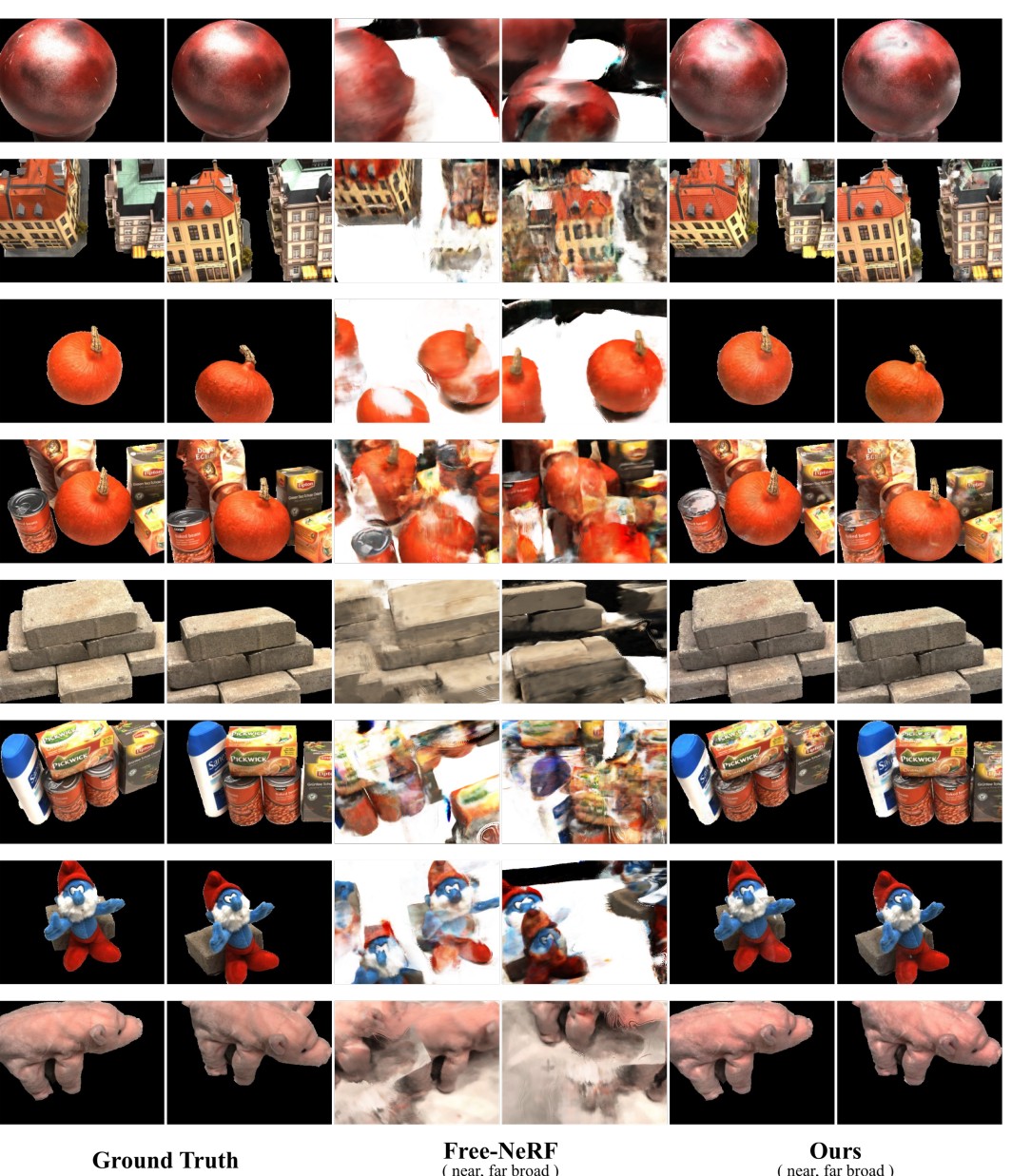

Ground Truth        **Free-NeRF**
( near, far broad )        **Ours**
( near, far broad )

Figure 10: **9-view result of DTU dataset.** Rendering result on the DTU dataset (9-view) comparing FreeNeRF and our method, with near and far values set to 0.05 and 1000, respectively.

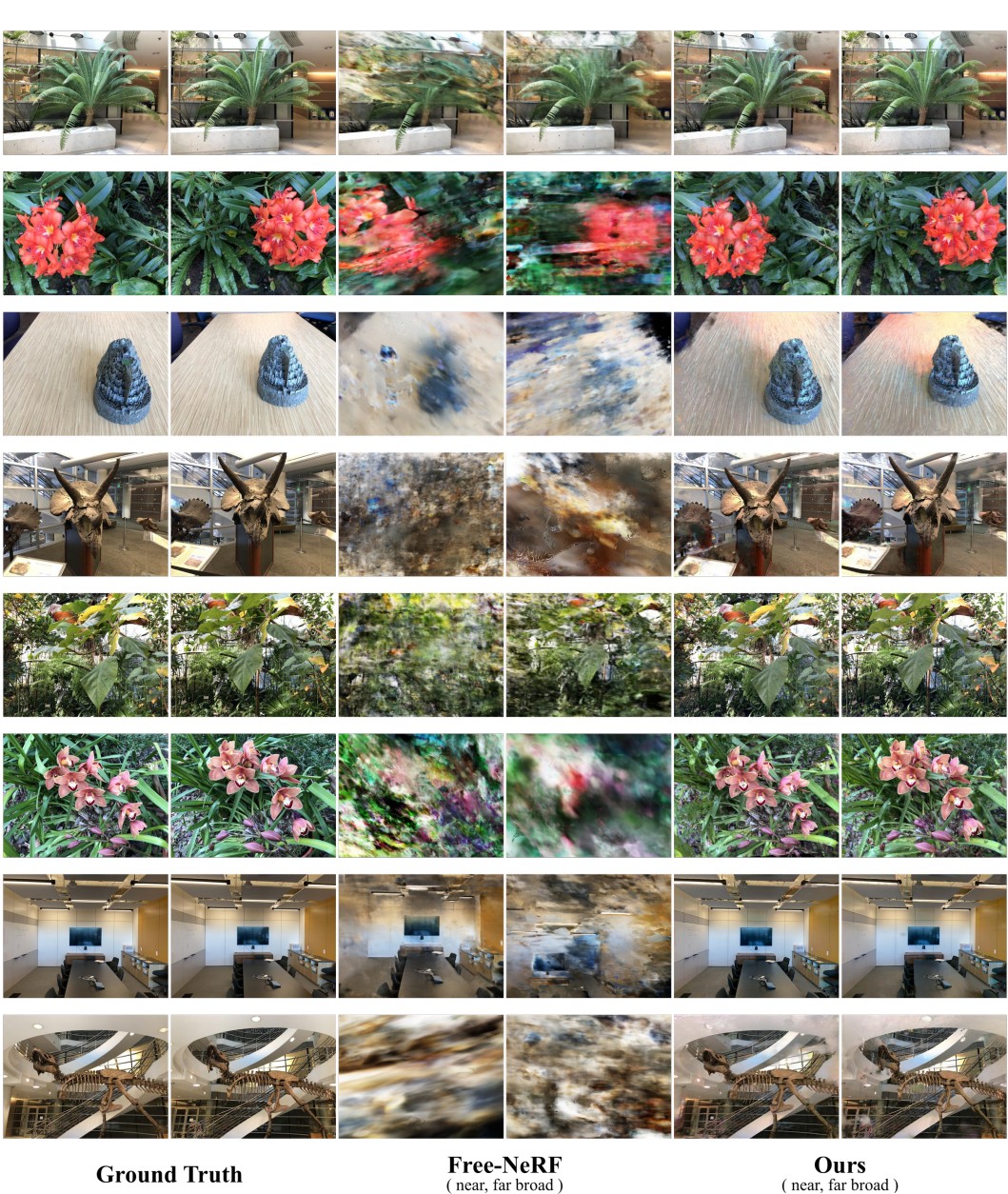

**Ground Truth**  **Free-NeRF**  **Ours**
( near, far broad )  ( near, far broad )

Figure 11: **6-view result of LLFF dataset.** Rendering result on the LLFF dataset (6-view) comparing FreeNeRF and our method, with near and far values set to 0.05 and 1000, respectively.

