# OpenReview forum: "Well-NeRF: Ensuring Well-Posed Neural Radiance Fields via View Frustum and Shadow Zone Based Regularization"
_ICLR.cc/2025/Conference — Submitted to ICLR 2025_

### Official Review · Reviewer_sTgH · 2024-10-29

**Soundness:** 3
**Presentation:** 2
**Contribution:** 2
**Rating:** 6
**Confidence:** 5

**Summary:**

This paper explores two key factors that lead to artifacts when training NeRF from sparse viewpoints. The first is that the regions cannot be inferred where the view frustum does not cover, and the second is that the opaque areas inside objects cannot be well constrained by the RGB loss, resulting in a NeRF that does not fully represent scene information. To address these issues, the paper proposes two regularization terms to constrain the training, focusing the network on areas with higher reliability for better results. Experiments demonstrate that the proposed method effectively resolves the two issues mentioned above.

**Strengths:**

The paper proposes two different strategies to address the underfitting issues caused by sparse viewpoints, which are actually quite common. These strategies involve designing regularization terms to constrain the training of NeRF, and have been proven to be very effective in experiments. On the other hand, the proposed approach is very intuitive and relatively concise, and it can achieve better results without altering other network settings. Additionally, from what I've observed, the current design is essentially a plug-and-play module that can be used in any existing NeRF. If possible, the author could also emphasize this point and validate it with some experiments.

**Weaknesses:**

For writing, firstly, regarding the paper's title, since the core problem addressed is how to regularize NeRF training under a "sparse perspective" to achieve better results, it's recommended that the title reflects the concept of "sparsity". Next, concerning the literature review section of the paper, I believe there should be relevant studies from the latest year (2024), and the author needs to thoroughly research the most recent advancements in this field. Additionally, there are issues with some of the formulas where the symbols are not clearly described, and there is inconsistency in their use. For example, the symbol (\sigma) in equation (3) should be consistent with equation (1). Also, it's unclear how (S_norm) in equation (6) is calculated—is it (S / num_views)?

For technical part, some design choices have not provided reasonable explanations for certain aspects.

In equation (3), the Frustum Score is a constant value for each sample point when the camera parameters are fixed. Therefore, the obtained \sigma_masked could be directly used in the integration calculation for RGB. Why then is there a need to further constrain its sparsity? Similarly, in equation (6), clipping is performed in the calculation of gradients. If, as previously mentioned, \sigma_masked is directly used in the integration, constraining both RGB and the gradients, wouldn't a similar effect be achieved with reduced computational effort? I hope the author can explain the design principles.

Additionally, during the RGB blending process, introducing RGB values near the surface into internal sample points through blending might cause color bleeding in other views. Is it also possible that the color of sample points after weighting could bring noise from unconstrained foreground areas into the interior?

Furthermore, in terms of experimental design, the paper only compares with FreeNeRF, but the settings of FreeNeRF are completely different from this study (network structure, use of hash acceleration, etc.), making the comparison potentially unfair. Lastly, I hope the author could add a few examples of novel view synthesis, because having constrained NeRF, theoretically, the results of NVS should improve. Otherwise, it might be possible that the training data was overfitted through regularization strategies.

**Questions:**

As discussed above.

---

> ### Author Response · Authors · 2024-11-22
> **Response to Reviewer sTgH**
>
> Thank you very much for investing your time in reviewing our work. Your detailed review is greatly appreciated and will be immensely helpful in improving our paper.
>
> **W1.**
>
> The reviewer's point is valid. We are currently considering changing the title.
>
> Our experiments were initially aimed at addressing the sparse input view problem. However, we realized that establishing reasonable spatial boundary conditions for training and resolving ill-posed problems carries a greater significance. **We believe this insight can be emphasized not only for sparse input conditions but also in addressing other challenges.**
>
> In fact, we demonstrated through simple yet important experiments that our implementation improves performance even under dense input conditions—an experiment not attempted by most other models addressing sparse input problems. Our model effectively preserves the performance of the original NeRF because, in the primary training regions, the influence of regularization is minimal, allowing it to operate identically to the original model.
>
> In contrast, other models addressing the sparse input problem often risk potential performance degradation due to excessive regularization in the regions of interest. For example, in the case of Free-NeRF, overfitting to low-frequency regions can result in reduced resolution.
>
> **W2.**
>
> We acknowledge the lack of references to and discussion of recent related research.
>
> Our work aimed to emphasize the potential of what is now considered a somewhat classic NeRF method. While remarkable recent models such as ZeroRF, Reconfusion, and Cat3D (as mentioned by other reviewers) exist, our approach was specifically designed to explore the potential of base models and their extensions built on the following foundations:
>
> - **Random pixel ray sampling**
> - **Encoding:** 3(position)+2(direction) input dimensions → n-dimensional output (including spatial scale levels)
>     - Examples:
>         - Frequency encoding
>         - Hash encoding
> - **Fully connected layers**
> - **Volume rendering**
>
> Due to this focus, the range of comparable models was significantly restricted to those sharing a similar architectural foundation.
>
> **W3.**
>
> We will work on improving the consistency of the notation.
>
> **W4.**
>
> S_adj = min(S, 9) - 1
>
> S_norm = S_adj / max(S_adj)
>
> We limited the Frustum Score count to 9 or less to prevent excessive gradient differences. When \( S = 1 \), position inference is impossible (as the point is observed from only one view). To eliminate training gradients in this case, we subtracted 1.
>
> This is explained in **Supplementary Material Equations 6 and 7**, but we noticed some notation inconsistencies in those equations. We will make the necessary corrections.
>
>
> **W5.**
>
> Q. In equation (3), the Frustum Score is a constant value for each sample point when the camera parameters are fixed. Therefore, the obtained \sigma_masked could be directly used in the integration calculation for RGB.
> ...
>
> We would appreciate it if you could confirm whether we have understood this question correctly. Thank you for thoroughly reviewing the model architecture.
>
> Our design principles were based on the following two assumptions, which we considered to be true:
>
> 1. The position of density and color cannot be inferred in regions where the view frustum does not intersect.
> 2. Information inside opaque surfaces cannot be observed or inferred, and thus cannot contribute to the rendering of the image.
> We attempted to implement these principles within the model as thoroughly as possible.
>
> Masking and gradient clipping can be applied to tensors with the shape [batch, num_sampling_points, n] that are generated during the computation process. For example:
>
> [batch, num_sampling_points, position(3)] (all corresponding tensors are aligned with this position):
>
> (a) [batch, num_sampling_points, density(1)]: gradient , mask
>
> (b) [batch, num_sampling_points, weight(1)]
>
> (c) [batch, num_sampling_points, rgb(3)]: gradient
>
> Integration of (weight * rgb) produces:
>
> [batch, rgb(3)]
>
> We believe the reviewer is suggesting that the method could be applied only at the step marked as the integration of (weight * rgb) to achieve the same effect.
>
> Additionally, we understand that the reviewer might be asking whether the method was redundantly applied to both (a) and (b).
>
> Our reasoning was as follows:
>
> During the computation of the weight, we believed that the density of inference-impossible regions could influence the weight of inference-possible regions, causing gradients to propagate into inference-impossible regions.
> Therefore, we applied the method again in step (a).
> While this implementation may not be a logically perfect equivalence to the first assumption, we believe that our experimental results demonstrate that the assumptions were convincingly implemented.

---

> ### Author Response · Authors · 2024-11-22
>
> **W6.**
>
> **Q.** Is it also possible that the color of sample points after weighting could bring noise from unconstrained foreground areas into the interior?
>
> The method is carefully designed to minimize its influence on regions where the base NeRF model can already determine results autonomously. Our approach aims only to assist NeRF in resolving the indeterminacy of internal colors that it cannot infer.
>
> This method does not affect other regions because the degree of color blending is proportional to the sampling point’s weight. As a result, color blending propagates unidirectionally from high-weight regions (e.g., opaque object surfaces) toward deeper areas along the viewing direction (inside the object). In low-weight regions, color blending has minimal impact on rendering, making its effect difficult to observe.
>
> The regularization coefficient $\lambda$ for Shadow Zone Regularization decreases significantly with iterations (as shown in Equation 10) to prevent overfitting. Additionally, the regularization loss $L_b$ (Equation 7) becomes much smaller than the primary training loss $L_c$ (Equation 8), helping the model converge during early training while having little to no impact in later stages.
>
> This method proved to be an effective approach among various experiments aimed at minimizing the indeterminacy of sampling points along a single ray.
>
> **W7.**
>
> **Q.** Settings of FreeNeRF are completely different from this study
>
> The frequency encoding used in Free-NeRF and hash encoding might seem significantly different at first glance, but the two methods share substantial similarities. While there are notable differences in the detailed implementations, both ultimately produce a multi-resolution (scale) position-encoded tensor that is concatenated. As a result, the structure of the final tensor is quite similar in both approaches.
>
> Thus, the frequency-level masking and scheduling implemented in Free-NeRF can be applied in a similar manner to hash encoding.
>
> In fact, we successfully implemented this and confirmed that it functions similarly. However, based on our independent implementation, we found that frequency masking is highly sensitive to scheduling and poses a significant risk of overfitting to low-frequency regions during the early stages, potentially resulting in reduced resolution.
>
> Nonetheless, we compared our method with the original implementation rather than relying solely on our independent implementation, as we believe a direct comparison would be more appropriate.
>
> It is somewhat unfortunate that we could not compare with more implementations, but we encountered a particular issue when attempting comparisons with other models.
>
> We attempted to compare our model with **Reg-NeRF**. However, we encountered difficulties due to the complex and ambiguously defined experimental conditions (particularly the near-far threshold). Based on our understanding, Reg-NeRF first applies a near-far condition, then performs coordinate warping into the NDC space, and subsequently applies a secondary near-far thresholding process within the NDC space. These settings are not only hardcoded but also vary numerically depending on the data class. Unfortunately, neither the main text nor the supplementary material of the Reg-NeRF paper explicitly specifies these conditions.
>
> As a result, we could not ensure that our comparison with Reg-NeRF was conducted under equal conditions and were therefore unable to include those results in our paper.
>
> This issue is not unique to Reg-NeRF but is common across many models, as they often fail to explicitly detail the hyperparameter settings (e.g., near-far thresholds) used in their codebases
>
> **W8.**
>
> We plan to provide experiments on new data and a wider range of views for existing experiments within the review period.
>
> We are conducting additional experiments, including addressing the points raised by you and other reviewers. Additional responses and materials will be uploaded during the review period, and we would greatly appreciate your continued interest.

---

> > ### Comment · Reviewer_sTgH · 2024-11-25
> > **Response to Authors**
> >
> > Thanks for your efforts and detailed explanations. The responses indeed resolve some of my concerns.
> >
> > What I mean in W5 is that, the Frustum Score remains constant when the camera parameters are set. Consequently, the obtained \sigma_masked is also constant. So, why is there a need to further constrain the sparsity of \sigma_masked,  will it change during training?
> >
> > Regarding W6, I remain unconvinced by the statement because the degree of color blending correlates directly with the weight of the sampling points. Could we potentially limit the color blending by setting a threshold to prevent color bleeding?
> >
> > Besides, as noted, comparing the proposed methods with others, such as Reg-NeRF, is challenging due to the complexity and ambiguous definition of the experimental conditions. I'm curious whether it's feasible to treat the two proposed regularizations as plug-and-play modules to see if they enhance performance beyond the original implementation.
> >
> > I am looking forward to the additional experiments. Thanks.

---

> > > ### Author Response · Authors · 2024-11-28
> > > **Response to Reviewer sTgH**
> > >
> > > Dear Reviewer,
> > >
> > > We have uploaded additional experiments and materials, with a summary provided in the comments at the top of this page.
> > >
> > > Among the uploaded materials, **2. Effectiveness of Dynamic Lambda** may be indirectly related to your concerns.
> > >
> > > Additionally, **4. New Dataset Proposal: Randomized Structures and Patterns** is proposed as a new contribution from our work.
> > >
> > > We are still preparing a direct response to your recent comments. We are carefully considering them and will provide a reply as soon as possible during the discussion period.
> > >
> > > Thank you for your time and understanding.

---

> ### Author Response · Authors · 2024-11-29
> **Official Comment by Authors**
>
> Thank you for taking the time to review our response. We sincerely appreciate your feedback and would be more than happy to address any remaining concerns you may have. Please do not hesitate to let us know if there is anything further we can do to fully address your concerns.

---

> ### Author Response · Authors · 2024-12-01
> **Response to Reviewer sTgH**
>
> ### Q1-1. The Frustum Score remains constant when the camera parameters are set.
> When the camera parameters are given, what is determined is the **Frustum Score Calculator**.
> However, the Frustum Score itself is recalculated for different sampling points in each iteration. Since the sampling points are determined by the proposal network during each iteration, the Frustum Score changes at every step.
>
> ### Q1-2. Consequently, the obtained $\sigma_{\text{masked}}$ is also constant.
> Because the sampling points vary with each iteration, $\sigma_{\text{masked}}$ also changes in every iteration.
>
> ### Q1-3. So, why is there a need to further constrain the sparsity of $\sigma_{\text{masked}}$? Will it change during training?
> While the **Frustum Score Calculator** is pre-defined and constant, the **Frustum Score** and $\sigma_{\text{masked}}$ are recalculated differently for each iteration.
>
> ### Summary of How Frustum Score is Used in the Model:
>
> 1. **Camera Parameters** → Pre-defined **Frustum Score Calculator (FSC)**.
> 2. **Ray Generator** → Proposal Network Sampler → Sampling Points Along the Ray (varies per iteration).
> 3. Frustum Score (FS) of sampling points is calculated using the FSC.
>    - At this stage, FS does not yet impose any constraints on training.
> 4. FS is then utilized as:
>    - **Frustum Score Mask (FSM)**.
>    - **Frustum Score Gradient Scaling (FSGS)**.
>
> These two components (FSM and FSGS) impose constraints on training.
>
> ### Q2. The degree of color blending correlates directly with the weight of the sampling points. Could we potentially limit the color blending by setting a threshold to prevent color bleeding?
>
> For sampling points along the ray, the color behaves as follows:
> 1. When the **weight** is small, the amount of blending is minimal.
> 2. When the **weight** is small, the contribution to rendering is minimal.
> 3. Even if some color bleeds slightly toward other input views, the model can recover as training progresses. This is because the regularization coefficient $\lambda \$ becomes very small over time. (Refer to Question 3 from Reviewer1 e43w and Section 2 of the Rebuttal for more details.)
>
> Setting a threshold is a great suggestion. However, in this model design, we have aimed to minimize the introduction of hyperparameters that require specific values. As such, we will consider whether additional designs can be introduced that minimize the risk of color bleeding outside the object without relying on fixed thresholds.
>
> ### Q3.
>
> Our module is designed as a plug-and-play module based on Nerfacto, allowing for easy integration with other methods. As a result, we have observed the outcomes of various methods operating under sparse conditions (refer to Figure 9 in the main text).
>
> It seems that the reviewer is asking whether other sparse models could see performance improvements when combined with our method in a plug-and-play manner. We will review whether this can be demonstrated quickly using our current implementation.
>
>
> **If there are any misunderstandings or differences in interpretation, please feel free to point them out. We are planning additional responses and would greatly appreciate your continued interest in our work.**

---

> > ### Comment · Reviewer_sTgH · 2024-12-03
> > **Response to Authors**
> >
> > Thanks for your efforts again. The additional experiments and explanation address most of my concerns. I will raise my rating to marginally above. Please include the necessary discussions in the revision, thanks.

---

> > > ### Author Response · Authors · 2024-12-03
> > > **Thank you for the positive feedback**
> > >
> > > Dear Reviewer,
> > >
> > > I am glad to hear that your concerns have been resolved. Your comments have been highly valuable in improving the soundness and clarity of the paper. I sincerely appreciate your positive evaluation.

---

### Official Review · Reviewer_xeNz · 2024-11-03

**Soundness:** 2
**Presentation:** 1
**Contribution:** 2
**Rating:** 3
**Confidence:** 4

**Summary:**

The work addresses NeRF reconstruction using sparse inputs. The authors assume that the primary cause of reconstruction artifacts is the incorrect setting of boundary conditions in the learning space. To tackle this issue, they propose a new regularization method based on two assumptions: (1) the position of density and color cannot be inferred in regions where the view frustum does not intersect, and (2) information inside opaque surfaces cannot be observed or inferred, and therefore cannot contribute to the rendering of the image. The proposed method can be seamlessly integrated with existing techniques.

**Strengths:**

1. The proposed regularization methods, along with the propositions to develop them, are logical and well-founded.
2. The proposed method automatically adjusts the training space without requiring additional parameter tuning, making it easy to combine with other approaches.

**Weaknesses:**

1. Regarding Shadow Zone Regularization, the authors claim that Equation 7 blends the opaque surface's color into the object's interior. However, there is no explicit determination of the opaque surface. How does this effect apply only to the interior of the object as claimed, without affecting other regions?
2. No video results are presented to demonstrate the reconstruction accuracy and view-consistency of the rendering. Additionally, the comparison baseline only involves nerfacto and FreeNeRF, which is insufficient.
3. Manually adjusting the bounding box is straightforward with popular NeRF frameworks and may achieve the same or even better results than the proposed Frustum Score Regularization.
4. Quotation marks are not used correctly. (Minor issue, not considered in my rating).

**Questions:**

1. Could the authors provide additional frustum score visualizations, especially those associated with the presented qualitative results? This would aid in understanding the proposed regularization.
2. Could the authors include loss curves for the proposed regularizers?
3. Could the authors also provide visualizations of RGB values of samples along the ray to illustrate the behavior of Shadow Zone Regularization?
4. Could the proposed approach be combined with 3D Gaussian Splatting? This is significant, as 3DGS is becoming the mainstream approach for novel view synthesis, surpassing NeRF.

---

> ### Author Response · Authors · 2024-11-22
> **Response to Reviewer xeNz**
>
> # **Response to Weaknesses**
>
> **W1. Regarding Shadow Zone Regularization, the authors claim that Equation 7 blends the opaque surface's color into the object's interior. However, there is no explicit determination of the opaque surface. How does this effect apply only to the interior of the object as claimed, without affecting other regions?**
>
> We encourage you to consider this method from the following perspective: rather than focusing on color blending, it seeks to minimally intervene in areas where the NeRF network cannot make independent decisions—such as unobserved regions or the interior of opaque surfaces. The method is carefully designed to minimize its influence on regions where the base NeRF model can already determine results autonomously.
>
> Decisions regarding opaque surfaces rely entirely on the capabilities of the original NeRF model. Our approach aims only to assist NeRF in resolving the indeterminacy of internal colors that it cannot infer.
>
> This method does not affect other regions because the degree of color blending is proportional to the sampling point’s weight. As a result, color blending propagates unidirectionally from high-weight regions (e.g., opaque object surfaces) toward deeper areas along the viewing direction (inside the object). In low-weight regions, color blending has minimal impact on rendering, making its effect difficult to observe.
>
> The regularization coefficient $\lambda$ for Shadow Zone Regularization decreases significantly with iterations (as shown in Equation 10) to prevent overfitting. Additionally, the regularization loss $L_b$ (Equation 7) becomes much smaller than the primary training loss $L_c$ (Equation 8), helping the model converge during early training while having little to no impact in later stages.
>
> This method proved to be an effective approach among various experiments aimed at minimizing the indeterminacy of sampling points along a single ray.
>
> **W2. No video results are presented to demonstrate the reconstruction accuracy and view-consistency of the rendering. Additionally, the comparison baseline only involves nerfacto and FreeNeRF, which is insufficient.**
>
> We will provide the video within the review period.
>
> We acknowledge the lack of comparisons with other methods and would like to provide additional context regarding our experimental settings.
>
> We attempted to compare our model with **Reg-NeRF**. However, we encountered difficulties due to the complex and ambiguously defined experimental conditions (particularly the near-far threshold). Based on our understanding, Reg-NeRF first applies a near-far condition, then performs coordinate warping into the NDC space, and subsequently applies a secondary near-far thresholding process within the NDC space. These settings are not only hardcoded but also vary numerically depending on the data class. Unfortunately, neither the main text nor the supplementary material of the Reg-NeRF paper explicitly specifies these conditions.
>
> As a result, we could not ensure that our comparison with Reg-NeRF was conducted under equal conditions and were therefore unable to include those results in our paper.
>
> This issue is not unique to Reg-NeRF but is common across many models, as they often fail to explicitly detail the hyperparameter settings (e.g., near-far thresholds) used in their codebases.
>
> While we recognize the limited comparisons as a weakness of our study, we conducted an in-depth review to address this and highlight that our model overcomes these issues by being a **near-far threshold-free model**. This characteristic sets our approach apart and addresses a major limitation observed in other models.
>
> **W3. Manually adjusting the bounding box is straightforward with popular NeRF frameworks and may achieve the same or even better results than the proposed Frustum Score Regularization.**
>
> Does the bounding box refer to limiting the training region using an axis-aligned bounding box (AABB) or an oriented bounding box (OBB)? Or does it refer to a precisely defined near-far threshold?
>
> Our work aims to establish clear decisions on spatial boundary conditions for training through theoretical calculations.
>
> Manually adjusting bounding boxes to achieve high performance would require iterative experiments. Our method minimizes this effort.
>
> The near-far range of 2-6 is a long-standing and reasonable setting for the Lego model. However, as shown in the middle row of **Figure 5**, this setting fails significantly under sparse input conditions (split ratio 0.03). The fragments around the object are caused by inference-impossible regions that are not adequately constrained by the near-far settings.
>
> Inference-impossible regions may also occur with AABB or OBB configurations. This is because the refined inference region calculated by our **Frustum Score Calculator** takes the form of a polyhedral polygon, which cannot be fully represented by a rectangular bounding box.

---

> ### Author Response · Authors · 2024-11-22
> **Response to Reviewer xeNz**
>
> # **Response to Questions**
>
> **W1. Could the authors provide additional frustum score visualizations, especially those associated with the presented qualitative results? This would aid in understanding the proposed regularization.**
>
> We plan to provide intuitive visualizations along with a video during the review period.
>
> **W2. Could the authors include loss curves for the proposed regularizers?**
>
> We will provide it within the review period.
>
> **W3. Could the authors also provide visualizations of RGB values of samples along the ray to illustrate the behavior of Shadow Zone Regularization?**
>
> We will provide it within the review period.
>
> **W4. Could the proposed approach be combined with 3D Gaussian Splatting? This is significant, as 3DGS is becoming the mainstream approach for novel view synthesis, surpassing NeRF.**
>
> Analyzing our method in comparison with Gaussian Splatting-based approaches is indeed an interesting task. However, there are important considerations to keep in mind. Most Gaussian Splatting-related works begin training with a point cloud obtained through SfM as a prior input.
>
> In contrast, many NeRF models do not use such inputs and start training solely with images and camera positions.
>
> Comparing Gaussian Splatting, which starts with 3D prior information about image surfaces, with NeRF models requires careful attention. It would be more appropriate to compare our method with Gaussian models that either use random Gaussian priors or no priors at all.
>
> Although NeRF and Gaussian Splatting share similarities, they differ significantly in their rendering methods (volume rendering vs. alpha-sorting and rasterization). Therefore, implementing such a comparison may take some time.
>
> We are primarily focused on maximizing the potential of NeRF and its implicit network capabilities. We also plan to design future projects to integrate our method with Gaussian Splatting.
>
> We are conducting additional experiments, including addressing the points raised by you and other reviewers. Additional responses and materials will be uploaded during the review period, and we would greatly appreciate your continued interest.

---

> > ### Author Response · Authors · 2024-11-28
> > **Response to Reviewer xeNz**
> >
> > Dear Reviewer,
> >
> > We have uploaded additional experiments and materials, with a summary provided in the comments at the top of this page.
> >
> > We carefully considered your comments and designed and conducted new experiments. In particular:
> >
> > **2. Effectiveness of Dynamic Lambda**
> >
> > **3. Visualization of Frustum Score**
> >
> > **5. Video Demonstration of NeRF Rendering and Loss Curve**
> >
> > are directly related to your feedback.
> >
> > Regarding your question:
> >
> > Q. How does this effect apply only to the interior of the object as claimed, without affecting other regions?
> >
> > We interpret this as a concern about potential over-regularization. We hope that Supplementary Material 2 addresses this concern to some extent. Additionally, the other experiments may also indirectly help alleviate your concerns.
> >
> > Thank you for your time and consideration.

---

> ### Author Response · Authors · 2024-11-29
> **We Look Forward to Your Valuable Feedback on Our Response**
>
> We look forward to receiving your valuable feedback on our response and would be more than happy to address any concerns you may have. Please do not hesitate to let us know if there is anything further we can do to address your concerns comprehensively.

---

> ### Comment · Reviewer_xeNz · 2024-12-03
>
> Dear authors,
>
> Thanks for your response, revision, and additional results. After checking them and other reviewers' opinions, I am still fully convinced of this work's contribution and would like to keep my original rating.
>
> As for the proposed shadow zone regularization, I still find it hard to operate only in the desired region. Moreover, I regard the efforts to adjust bounding box affordable, given the current speed of NeRF training. So I don't think this work brings a significant improvement to the community.
>
> The new results demonstrate improvement over nerfacto, but there is still a lack of comparison with other sota methods. Even the nerfacto results can be improved according to my experience.

---

> ### Author Response · Authors · 2024-12-03
> **Thank you for the feedback**
>
> Dear Reviewer,
>
> Thank you very much for your response. We deeply respect your evaluation.
>
> To elaborate further on some points:
>
> **W1.** As for the proposed shadow zone regularization, I still find it hard to operate only in the desired region.
>
> This method was designed based on an assumption of intuition regarding rigid surfaces and rendering, which may raise concerns that it might not function perfectly as intended in all cases. However, we believe that the design already incorporates considerations to address such concerns (e.g., preventing over-regularization), and the performance improvements were explicitly demonstrated through the ablation study.
>
> **W2.** Moreover, I regard the efforts to adjust bounding box affordable, given the current speed of NeRF training.
>
> It is often difficult to achieve good results even with the best possible bounding box because a complex, polygon-shaped learnable region cannot be adequately represented by a simple bounding box.
>
> In practical scenarios where NeRF is applied as a service, it is hard to expect general users (non-specialists or researchers) to conduct repeated experiments.
>
> I view the proper restriction of the learning region not merely as an extension of parameter adjustment or 3D editing but as a theoretical contribution that makes NeRF a well-posed problem, akin to a differential equation with appropriately defined boundary conditions.
>
> **W3.** but there is still a lack of comparison with other sota methods.
>
> It is indeed unfortunate that our model does not actively compete with other models in terms of SOTA performance. However, we highlighted in our work that over-regularization methods or optimization issues for specific data classes have arisen in pursuit of SOTA performance in NeRF. We attributed these issues to a lack of dataset diversity. To address this, we proposed a completely new dataset, and we hope this will be considered an additional contribution (please refer to Section 4 of our rebuttal: New Dataset Proposal: Randomized Structures and Patterns).
>
> We accept your evaluation and want to express that the discussion process has been instrumental in improving the paper. We will continue to reference this feedback in the future. Thank you again for your valuable input.

---

### Official Review · Reviewer_M1eU · 2024-11-03

**Soundness:** 3
**Presentation:** 3
**Contribution:** 2
**Rating:** 5
**Confidence:** 5

**Summary:**

This paper proposes, Well-NeRF, which leverages view frustum and shadow zone-based regularization to make NeRF a well-posed problem under sparse view setting. Authors show outstanding performance across various test datasets.

**Strengths:**

+: The paper introduces the idea of addressing sparse input issues through frustum score and shadow zone regularization, which is highly simple and reasonable.
+: The experimental results are impressive, across different scenes.
+: The analysis is fundamental in 3D reconstruction community, which may inspire other 3D research task.

**Weaknesses:**

-: The dataset are too small to well demonstrate the upper bound of the proposed method. And then the results are sensitive to the experimental setting, making the results less convincing.
-: Lack enough comparisons to other methods, like RegNeRF[1], ZeroRF[2], etc.
-: Lack experiments on various number of views, making me unclear about the sensitivity and scalability.
-: (not totally a weakness, but a suggestion) The whole analysis seems independent of how we represent the scene. So, why not enhance the paper with experiments on 3DGS?

[1] Regnerf: Regularizing neural radiance fields for view synthesis from sparse inputs
[2] ZeroRF: Fast Sparse View 360◦ Reconstruction with Zero Pretraining

**Questions:**

Please refer to the weakness.

---

> ### Author Response · Authors · 2024-11-22
> **Response to Reviewer M1eU**
>
> ## Response to Weaknesses and Questions
>
> **Q1. The dataset are too small to well demonstrate the upper bound of the proposed method**
>
> We are planning experiments on new datasets.
>
> However, we would like to point out that there is currently a lack of suitable datasets for few-shot experiments. When splitting data into training and evaluation sets, there is a high likelihood that unobserved regions in the training set are included in the evaluation set. In such scenarios, how should the model approximate completely unobserved regions? As black, white, gray, random colors, or the mean color? This is nonsensical and sometimes leads researchers to enforce overfitting to achieve high evaluation PSNR.
>
> Therefore, we are designing a random structure, random texture dataset and plan to create hundreds to thousands of samples. This approach aims to eliminate potential biases in evaluations.
>
> **Q2. And then the results are sensitive to the experimental setting, making the results less convincing.**
>
>  Our goal is to ensure functionality across a wide range of near and far plane settings, from narrow to wide configurations. While the ultimate aim is to make the method near-far parameter-free, achieving exact values of 0 and infinity is numerically impossible. Therefore, we used a setting of 0.5 to 1000. Could you clarify which aspects you found to be sensitive to the experimental settings?
>
> **Q3. Lack enough comparisons to other methods, like RegNeRF, ZeroRF, etc.**
>
> We acknowledge the lack of comparisons with other methods and would like to provide additional context regarding our experimental settings.
>
> We attempted to compare our model with **Reg-NeRF**. However, we encountered difficulties due to the complex and ambiguously defined experimental conditions (particularly the near-far threshold). Based on our understanding, Reg-NeRF first applies a near-far condition, then performs coordinate warping into the NDC space, and subsequently applies a secondary near-far thresholding process within the NDC space. These settings are not only hardcoded but also vary numerically depending on the data class. Unfortunately, neither the main text nor the supplementary material of the Reg-NeRF paper explicitly specifies these conditions.
>
> As a result, we could not ensure that our comparison with Reg-NeRF was conducted under equal conditions and were therefore unable to include those results in our paper.
>
> This issue is not unique to Reg-NeRF but is common across many models, as they often fail to explicitly detail the hyperparameter settings (e.g., near-far thresholds) used in their codebases.
>
> While we recognize the limited comparisons as a weakness of our study, we conducted an in-depth review to address this and highlight that our model overcomes these issues by being a **near-far threshold-free model**. This characteristic sets our approach apart and addresses a major limitation observed in other models.
>
> **Q4. Lack experiments on various number of views, making me unclear about the sensitivity and scalability**
>
> We are currently conducting experiments to provide results for a wider range of views.
>
> **Q6. So, why not enhance the paper with experiments on 3DGS?**
>
> Analyzing our method in comparison with Gaussian Splatting-based approaches is indeed an interesting task. However, there are important considerations to keep in mind. Most Gaussian Splatting-related works begin training with a point cloud obtained through SfM as a prior input.
>
> In contrast, many NeRF models do not use such inputs and start training solely with images and camera positions.
>
> Comparing Gaussian Splatting, which starts with 3D prior information about image surfaces, with NeRF models requires careful attention. It would be more appropriate to compare our method with Gaussian models that either use random Gaussian priors or no priors at all.
>
> Although NeRF and Gaussian Splatting share similarities, they differ significantly in their rendering methods (volume rendering vs. alpha-sorting and rasterization). Therefore, implementing such a comparison may take some time.
>
> We are primarily focused on maximizing the potential of NeRF and its implicit network capabilities. We also plan to design future projects to integrate our method with Gaussian Splatting.
>
> We are conducting additional experiments, including addressing the points raised by you and other reviewers. Additional responses and materials will be uploaded during the review period, and we would greatly appreciate your continued interest.

---

> > ### Author Response · Authors · 2024-11-28
> > **Response to Reviewer M1eU**
> >
> > Dear Reviewer,
> >
> > Additional experiments and materials have been uploaded, with a summary provided in the comments at the top of this page.
> >
> > You identified the lack of experiments as a key weakness of our paper. In response, the additional materials include extended experiments based on the original ones as well as a proposal for an entirely new dataset accompanied by additional experiments. We believe these additions may increase the academic contribution of our work, and we kindly request your review.
> >
> > In particular, we invite you to review **4. New Dataset Proposal: Randomized Structures and Patterns.**
> >
> > Thank you for your time and consideration.

---

> ### Author Response · Authors · 2024-11-29
> **We Look Forward to Your Valuable Feedback on Our Response**
>
> We look forward to receiving your valuable feedback on our response and would be more than happy to address any concerns you may have. Please do not hesitate to let us know if there is anything further we can do to address your concerns comprehensively.

---

> ### Author Response · Authors · 2024-12-03
> **Reminder**
>
> The discussion period is coming to an end. We are looking forward to your valuable feedback. Thank you.

---

### Official Review · Reviewer_sTD1 · 2024-11-04

**Soundness:** 3
**Presentation:** 3
**Contribution:** 3
**Rating:** 6
**Confidence:** 5

**Summary:**

This paper proposes a simple method to improve nerf results under sparse input views (as few as 6). This method captures two intuitions: (1) points that only appear in just one input view are not learnable because the mass to learn can lie anywhere along the rays (paired with a specific scale) to show up correctly in the camera, and (2) points inside the object are not learnable since they don’t contribute to the final observed colors. The authors argue that explicitly making the network not learn those points improves the sparse-view performance.

For (1), the authors compute a mask indicating the times each point is included in all input views’ frustums, and use it to mask the loss and scale the gradients. For (2), the authors encourage nearby points to have similar colors, by gradually blending each point’s RGB with the previous point along the ray and also computing loss between the current RGB and the blended RGB.

The method is compared against nerfacto as a plug-and-play module, and the results show nerf results get improved in general with this trick. It also gets compared with FreeNerf, where near and far planes need selecting carefully when input views are sparse; this limitation can be eliminated with this paper, and similar results can be acheived.

The authors also test their method on real-world datasets including the well-known “nerf datasets” and a real in-the-wild dataset captured by the authors themselves (in the supplemental PDF).

**Strengths:**

This paper is strong in how it turns simple observations/intuitions into concrete implementations that improve nerf results in general. The intuitions make sense, and the end results indeed look improved.

The paper presents the ideas clearly with helpful visuals such as Figures 1 and 3.

The experiments from baseline comparisons to ablation studies are extensive and cover questions people may have very well.

**Weaknesses:**

I like the simplicity and modularity of this approach but the real-world, in-the-wild results shown in the supplemental material PDF are of concerning quality. Admittedly, high-quality view synthesis from just 6 input views of an in-the-wild scene is hard, but the method is shown to work well for the famous real-world “nerf datasets”. Clearly a gap here, one that needs closing before this approach is useful for any real use case.

A bigger question follows: Under sparse input views, do such per-scene learning approaches still make sense? I think when input views are sparse like this, and the quality presented is bad like this, one may be better off with learning-based approaches that learn from many scenes and generalize reasonably to the test scene at hand.

**Questions:**

Related to my point above about learning-based approaches that learn from multiple scenes, have the authors compared this approach against those approaches. There’s PixelNerf, and many nice works that followed. My intuition is the fewer input views you have, the better-suited a learning-based approach becomes, with priors learned from many scenes.

---

> ### Author Response · Authors · 2024-11-22
> **Response to Reviewer sTD1**
>
> We sincerely thank the reviewer for taking the time to review our work. All comments will be invaluable in improving the paper.
>
> ## Response to Weaknesses and Questions
>
> The reviewer's concerns are valid.
>
> The data from the Custom Indoor Dataset included in the Supplementary Material was provided to support our research motivation. Our initial challenge targeted the most complex (indoor) datasets. However, for step-by-step resolution, this paper simplifies the problem to identify one of the key reasons why many NeRF models struggle under sparse input conditions.
>
> ### Q: Under sparse input views, do such per-scene learning approaches still make sense?
>
> **A:** Yes, they are meaningful.
>
> For instance, if a spherical object with a well-defined surface pattern is observed under ideal conditions, its surface can be fully reconstructed through stereo vision using only four views. These views are positioned at 90-degree angles relative to a plane passing through the sphere’s center. (Each surface point is observed from at least two views.)
>
> Then why do NeRF models fail critically under sparse input view conditions? This issue can be better understood through the following analogy: NeRF models cannot distinguish between a scenario where a large sphere is centered in the scene and another where four small spheres obscure the camera directly in front of it. (The fact that classical models perform well under sparse conditions while deep learning models fail suggests that existing NeRF models may not yet be fully optimized.)
>
> We believe this failure stems from improperly defined training conditions. To address this, we implemented our model based on the following assumption:
>
> **Assumption 1:** *It is impossible to infer position at points where view frustums do not intersect (as triangulation is not feasible).*
>
> **Implementation:** The model was trained with boundary conditions set to restrict learning to regions where position inference is possible.
>
> Our proposed method is designed to influence the model only during the early iterations of training (as shown in Equation 10). After this initial phase, it functions almost identically to the base NeRF model (nerfacto).
>
> **This suggests that the potential of existing NeRF models remains significant.**
>
> ### Insights on NeRF and Generative Models
>
> We believe that the potential of existing NeRF models remains significant. NeRF is a technique for reconstructing the 3D structure and optical properties of a scene, replacing certain aspects of traditional surveying and graphics techniques with deep neural networks while maintaining foundational principles (e.g., volume rendering and optical physics). This highlights that NeRF serves as an extension rather than a complete replacement of conventional methods.
>
> Generative models excel at supplementing data or producing visually realistic results, which can enhance NeRF’s performance. However, generative models prioritize visual plausibility over metric accuracy, differing from NeRF in both approach and objectives.
>
> Therefore, as these technologies evolve and integrate, they can achieve optimal results. However, the combination of generative models does not render the development of NeRF-based models unnecessary, nor can it completely replace them.
>
> We refer to Reconfusion, one of the most recent generative models, for reference. In the initial video provided (https://reconfusion.github.io/videos/teaser/wipe_4.mp4), a phenomenon can be observed in Zip-NeRF where the object is not centered but instead rotates as if it were part of the background. Such object collapse is a common issue found in many models under sparse input conditions. This phenomenon is identical to the issue caused by incorrectly defined boundary conditions, as described in our paper.
>
> The collapse of the central object into the background can also be observed in Figure 8 of our main text.
>
> While Reconfusion has achieved remarkable success, we deliberately excluded pre-trained or generative NeRF models from the scope of this paper to avoid diluting our focus, especially considering the significant resources required for pre-training and the need for continuous refinement of base models. Furthermore, we believe that combining NeRF-based methods with generative models to address sparse input conditions is not yet essential, given the potential inherent in existing NeRF technologies.
>
> We are conducting additional experiments, including those addressing points raised by other reviewers. Additional responses and materials will be uploaded during the review period, and we appreciate your continued interest.

---

> > ### Author Response · Authors · 2024-11-28
> > **Response to Reviewer sTD1**
> >
> > Dear Reviewer,
> >
> > Additional experiments and materials have been uploaded, and a summary is provided in the comments at the top of this page. While there are no materials directly requested by you, we believe the additional data may help in revisiting the paper.
> >
> > In particular:
> >
> > **2. Effectiveness of Dynamic Lambda**
> >
> > **4. New Dataset Proposal: Randomized Structures and Patterns**
> >
> > are contributions that we believe further enhance our work.
> >
> > Thank you for your time and consideration.

---

> > > ### Comment · Reviewer_sTD1 · 2024-11-29
> > >
> > > Thanks for the authors' rebuttal, which provided some nice discussions between classical approaches and more recent learning-based approaches. I'm raising my rating to marginally above, and I hope the authors will include such discussions in the future version of their paper.

---

> > > > ### Author Response · Authors · 2024-12-03
> > > > **Thank you for the positive feedback**
> > > >
> > > > Dear Reviewer,
> > > >
> > > > The discussion with you has also been highly valuable to us. As you suggested, we are maintaining a continuous interest in the learning-based approach, and we plan to make efforts on both sides along with improving the base model. We sincerely appreciate your positive evaluation.

---

### Official Review · Reviewer_e43w · 2024-11-06

**Soundness:** 2
**Presentation:** 2
**Contribution:** 2
**Rating:** 6
**Confidence:** 3

**Summary:**

This paper introduces Well-NeRF, a method addressing sparse input problems in NeRF models. The proposed approach includes Frustum Score and Shadow Zone to constrain learning to well-posed regions in order to reduce artifacts. Experimental results on synthetic and real-world datasets demonstrate the method’s improvement over traditional models.

**Strengths:**

1. The two key assumptions are insightful and crucial.

2. Authors did well in the organization of the paper.

3. The idea of exploring view frustum seems interesting. It would be a great point for solving problems of sparse inputs in NeRFs.

**Weaknesses:**

1. **Incremental Contribution**. The authors did not give enough theoretical proofs and arguments for effectiveness of their method. In the Experiment part, as the experiments largely focus on synthetic data, the contribution of this work seems incremental. Please see Question 2 for more information on this weakness.

2. **The proposed method**. Although the authors give a good and novel assumption, the proposed methods seems simple and incremental without insightful design. More validation can contribute to the soundness of your method.

3. **Insufficient Experiments**. The authors did not provide sufficient experiment about comparisons with prior works to show their performances. The experiments are mainly conducted on **NeRF Synthetic Dataset**. However, the huge amount of experiments on synthetic datasets would weaken the effectiveness of the proposed methods for real world applications. I would encourage authors to conduct more experiment to improve the soundness of the paper.

4. **Writing**. Writing could be improved for clarity and soundness. Some typos and mistakes could be corrected i.e.  L45, 47, 50  incorrect quotation marks. L406 the sentence is not clear.

5. **Figures**. More detailed figures can improve the clarity of the paper and make your paper more understandable. Figures in the submission seem too simple for reader to fully understand your methods and arguments.

**Questions:**

1. Can you give the **Frustum Score** of the input views in your training settings? Also can you give a statement with **Frustum Score** to explain what kind of inputs improves the most with your method?
2. Can you give more experiments on near/far plane setting? Is the proposed method still work well or does it still improve over baselines with near/far settings other than [0.05,1000] (such as [2,6] in Figure. 5, 6)?
3. Can you provide ablation studies on lambda of Equation. 9? You can set lambda as a pre-set hyperparameter which do not change in the training. It seems interesting to detach part of the loss as another parameter. How much does this design contribute to the convergence speed?
4. Can you give the ablation studies on the number of sampling points? In Equation. 7, it seems that the parameters depend on the number of sampling points. Does the number of sampling points influence the training results?
5. Can you give more results on large-scale datasets? The datasets with sparse inputs can further demonstrate the efficacy of your method.
6. Can you provide the experimental results on comparisons with similar works with Gaussian splatting? If the results still improve greatly compared with them, it would greatly improve the performances of the work.

---

> ### Author Response · Authors · 2024-11-21
> **Response to Reviewer e43w**
>
> First of all, we sincerely thank you for taking the time to review and evaluate our paper. Your comments have provided valuable insights and have been greatly helpful in revisiting and improving our work.
>
> ## Response to Weaknesses
>
> The reviewer's point is valid.
>
> To provide additional clarification, the proposed method starts from two true propositions:
>
> **1. It is impossible to infer position at points where view frustums do not intersect (as triangulation is not feasible).**
>
> **2. Information inside solid objects cannot contribute to rendering (as it is unobservable).**
>
> Our implementation, inspired by these propositions, cannot be claimed as a logically perfect equivalence to them. However, through careful examination, we have made significant efforts to ensure these propositions are well reflected in our learning model, and experiments have demonstrated their effectiveness.
>
> The motivation for our challenge is as follows:
>
> The base model of **nefstudio** (https://github.com/nerfstudio-project/nerfstudio/), **nerfacto**, is one of the most widely used general-purpose models. While this model has been refined over a long time to achieve generality in real-world applications, users have occasionally faced difficulties training it on simple objects (examples below):
>
> - https://github.com/nerfstudio-project/nerfstudio/issues/2443
> - https://github.com/nerfstudio-project/nerfstudio/issues/806
>
> This issue remains unresolved and becomes especially pronounced when data is scarce. It has posed significant challenges in one of our recent practical projects.
>
> However, we assumed that if there is a spherical object with a well-defined surface pattern, under ideal conditions, the surface of the sphere could be fully reconstructed through stereo vision using only four views positioned to observe the sphere at 90-degree angles relative to a plane passing through the center of the sphere. (All surfaces can be observed from at least two of these views.)
>
> Nevertheless, many NeRF models fail critically in this scenario. We were convinced that the primary cause of this issue is related to the first assumption. (This problem can be better understood through the following analogy: a computer cannot distinguish between a scenario where a large sphere is at the center and another where four small spheres obstruct the camera directly in front of it.)
>
> Based on multiple reported cases and our own experience, we believed that our attempt and implementation could significantly contribute to improving the practical generalization of models. This implementation was achieved by introducing minimal modifications and module insertions into the nerfacto model, ensuring that the original principles of the model remain largely intact. As a result, it integrates seamlessly with all functionalities of **nerfacto** that can be toggled on or off.
>
> Our Appendix (at the end of the main paper) includes experiments on DTU and LLFF datasets. These datasets consist of real-world captured data, which we believe demonstrates the applicability of our method in real-world scenarios.
>
> We will work on improving the writing and figures. Additional responses regarding the experiments will follow in the next comments.

---

> ### Author Response · Authors · 2024-11-21
> **Response to Reviewer e43w**
>
> ## Response to Questions
>
> **Response to the Q1:**
>
> 1. If the input data includes camera intrinsics and extrinsics (the basic input format for NeRF training), the Frustum Score calculator matrix is precomputed. During training, the Frustum Score Calculator is used to calculate the scores of sampling points. Figure 3(d) represents the Frustum Score for actual sampling points calculated with three input views (yellow: 3 points, red: 2 points, yellow: 3 points).
> 2. So far, our experiments have shown that the method works effectively on object-centric synthetic datasets (Blender), forward-facing real-world datasets (LLFF), and real-world datasets captured along a quarter-sphere trajectory (DTU).
>
> **Response to the Q2:**
>
> The experiment is feasible and still works well. Our goal is to ensure functionality across a wide range of near and far plane settings, from narrow to wide configurations. While the ultimate aim is to make the method near-far parameter-free, achieving exact values of 0 and infinity is numerically impossible. Therefore, we used a setting of 0.5 to 1000.
>
> **Response to the Q3:**
>
> A controlled study on lambda is feasible. Our methodology is designed to minimize excessive regularization and maximize the potential of the model itself. It influences the model only during the very early stages, as lambda decreases in proportion to the RGB loss (and is further reduced by the overall optimization scheduler). Consequently, after the initial iterations, the model operates no differently from the original nerfacto.
>
> **Response to the Q4:**
>
> A controlled study on the number of sampling points is feasible. However, the loss does not increase proportionally to the number of sampling points. This is because we used PyTorch's default MSELoss with mean reduction (dividing by the number of elements). We will add a clarification about the exact formulation of MSELoss.
>
> On the other hand, adjusting the number of sampling points is a significant factor that affects both the original nerfacto model and the baseline NeRF results. Therefore, such a controlled study might introduce some confusion.
>
> **Response to the Q5:**
>
> We are planning experiments on new datasets.
>
> However, we would like to point out that there is currently a lack of suitable datasets for few-shot experiments. When splitting data into training and evaluation sets, there is a high likelihood that unobserved regions in the training set are included in the evaluation set. In such scenarios, how should the model approximate completely unobserved regions? As black, white, gray, random colors, or the mean color? This is nonsensical and sometimes leads researchers to enforce overfitting to achieve high evaluation PSNR.
>
> Therefore, we are designing a random structure, random texture dataset and plan to create hundreds to thousands of samples. This approach aims to eliminate potential biases in evaluations.
>
> **Response to the Q6:**
>
> Analyzing our method in comparison with Gaussian Splatting-based approaches is indeed an interesting task. However, there are important considerations to keep in mind. Most Gaussian Splatting-related works begin training with a point cloud obtained through SfM as a prior input.
>
> In contrast, many NeRF models do not use such inputs and start training solely with images and camera positions.
>
> Comparing Gaussian Splatting, which starts with 3D prior information about image surfaces, with NeRF models requires careful attention. It would be more appropriate to compare our method with Gaussian models that either use random Gaussian priors or no priors at all.
>
> Although NeRF and Gaussian Splatting share similarities, they differ significantly in their rendering methods (volume rendering vs. alpha-sorting and rasterization). Therefore, implementing such a comparison may take some time.
>
> **We plan to conduct the suggested experiments within the review period and provide additional responses. Thank you for your continued interest.**

---

> > ### Comment · Reviewer_e43w · 2024-11-21
> > **Response to Authors**
> >
> > Thanks for your reponse and efforts. After reading your response, I better understand your work and your contribution. I am looking forward to your additional responses and results of experiments.

---

> > > ### Author Response · Authors · 2024-11-28
> > > **Response to Reviewer e43w**
> > >
> > > Dear Reviewer,
> > >
> > > We have uploaded additional experiments and materials. A summary of these updates can be found in the comments at the top of this page. Among the uploaded materials:
> > >
> > > 1. Near-Far Parameter Robustness
> > > 2. Effectiveness of Dynamic Lambda
> > > 3. Visualization of Frustum Score
> > > 4. New Dataset Proposal: Randomized Structures and Patterns
> > >
> > > are particularly relevant to your comments.
> > >
> > > In particular, based on your feedback, we would like to emphasize Dynamic Lambda as another key contribution of our work.
> > >
> > > Thank you for your time, and we kindly request your review of the updates.

---

> > > > ### Comment · Reviewer_e43w · 2024-11-28
> > > > **Response to Authors**
> > > >
> > > > Thanks for your reply and efforts. Your response and additional experiments address most of my concerns. Therefore, I am raising my rating. Please add these additional experiments and results in the future versions to ensure the soundness and clarity of the paper.

---

> > > > > ### Author Response · Authors · 2024-12-03
> > > > > **Thank you for the positive feedback**
> > > > >
> > > > > Dear Reviewer,
> > > > >
> > > > > I am glad to hear that your concerns have been resolved. Your comments have greatly contributed to improving the paper. I sincerely appreciate your positive evaluation.

---

### Author Response · Authors · 2024-11-28
**Additional Materials for Rebuttal Uploaded**

Dear Reviewers,

Once again, we would like to sincerely thank you for investing your time in reviewing our paper. Your in-depth feedback has been incredibly valuable in providing insights and improving our work.

As promised, we have compiled and uploaded the requested additional experiments and supplementary materials. The files added during the rebuttal process can be found in the Supplementary Material.zip file, which includes **ICLR_2025_Well_NeRF_Supplementary Materials for Rebuttal.pdf** and the **Videos folder.**

The contents of **Supplementary Materials for Rebuttal** can be summarized as follows:

**1. Near-Far Parameter Robustness**

This experiment demonstrates the robustness of our model to near-far parameters. While the near-far parameter may seem simple, it is, in fact, a highly sensitive parameter that significantly impacts the management of the training region in existing models.

**2. Effectiveness of Dynamic Lambda**

The **Dynamic Lambda** we designed through experiments duplicates the RGB loss value and uses it to regulate the normalization lambda. This approach prevents the risk of over-regularization and enhances training stability. Additionally, during the mid-to-late stages of training, it minimizes the impact of regularization, allowing the base model to dominate. This ensures that the potential of the base model is preserved and highlighted. We would like to emphasize this as another key contribution of our work.

**3. Visualization of Frustum Score**

To observe how the view frustum score influences the training region, we visualized it alongside NeRF rendering during the training process.

**4. New Dataset Proposal: Randomized Structures and Patterns**

We previously mentioned to some reviewers that "there is currently a lack of reasonable datasets for NeRF experiments." In response, we propose a newly developed dataset generator. This generator creates datasets that are infinitely randomized, preventing experimenters from designing biased methods due to insufficient datasets. We hope this contributes to a more accurate evaluation of NeRF model designs. The dataset generator will be made publicly available for everyone to use.

**5. Video Demonstration of NeRF Rendering and Loss Curve**

We included video materials alongside the loss curve to provide a more detailed examination of our model's performance.

If the feedback is positive, the rebuttal materials will be incorporated into the paper and the Supplementary Materials.

Thank you.

---

### Meta-Review · Area_Chair_YAPV · 2024-12-18

**Metareview:**

In this paper, the authors have proposed Well-NeRF for reconstruction with sparse inputs. The motivation of this paper mainly comes from: 1) the points should not be able to inferred if they only appear in one view, and 2) the points inside the object are not observed so that should not contribute to the rendering process. By designing new regularization methods based on frustum score and shadow zone, Well-NeRF better restricts the boundary condition. The sparse view reconstruction is an important problem, and the proposed method should have potential to combine with more NeRF methods as a plug-and-play module. However, there are still some significant limitations of this paper. About the experiments, I suggest to apply the proposed method to more NeRFs and compare with more sota methods to better demonstrate the advantages of Well-NeRF. Reviewer xeNz raises concerns on the shadow zone regularization of the method. The writing of the paper should also be futher improved as mentioned by many reviewers. Based on the concerns above, I recommend a decision of rejection of this paper.

**Additional Comments On Reviewer Discussion:**

Initially the reviewers raised concerns on the writing, the significance of the setting, contributions, several technical details and experiments. In the rebuttal, the authors have addressed many of them, but there are still some issues remained as I mention in the metareview, which are important to make a final decision.

---

### Decision · Program_Chairs · 2025-01-22

Reject